# WEIGHT-SPACE SYMMETRY IN NEURAL NETWORK LOSS LANDSCAPES REVISITED

## ABSTRACT

Neural network training depends on the structure of the underlying loss landscape, i.e. local minima, saddle points, flat plateaus, and loss barriers. In relation to the structure of the landscape, we study the permutation symmetry of neurons in each layer of a deep neural network, which gives rise not only to multiple equivalent global minima of the loss function but also to critical points in between partner minima. In a network of $d-1$ hidden layers with $n_k$ neurons in layers $k = 1, \ldots, d$, we construct continuous paths between equivalent global minima that lead through a 'permutation point' where the input and output weight vectors of two neurons in the same hidden layer $k$ collide and interchange. We show that such permutation points are critical points which lie inside high-dimensional subspaces of equal loss, contributing to the global flatness of the landscape. We also find that a permutation point for the exchange of neurons $i$ and $j$ transits into a flat high-dimensional plateau that enables all $n_k!$ permutations of neurons in a given layer $k$ at the same loss value. Moreover, we introduce higher-order permutation points by exploiting the hierarchical structure in the loss landscapes of neural networks, and find that the number of $K$-th order permutation points is much larger than the (already huge) number of equivalent global minima – at least by a polynomial factor of order $K$. In two tasks, we demonstrate numerically with our path finding method that continuous paths between partner minima exist: first, in a toy network with a single hidden layer on a function approximation task and, second, in a multilayer network on the MNIST task. Our geometric approach yields a lower bound on the number of critical points generated by weight-space symmetries and provides a simple intuitive link between previous theoretical results and numerical observations.

## 1 INTRODUCTION

The structure of the loss landscape plays an important role in the optimization of neural network parameters. A large number of numerical (Dauphin et al., 2014; Goodfellow et al., 2014; Li et al., 2018; Sagun et al., 2014; 2016; Ballard et al., 2017; Garipov et al., 2018; Draxler et al., 2018; Sagun et al., 2017; Baity-Jesi et al., 2018) and theoretical (Choromanska et al., 2015; Rasmussen, 2003; Freeman and Bruna, 2016; Soudry and Carmon, 2016; Nguyen and Hein, 2017) studies have explored the properties of the loss landscape. In particular, in a multilayer network of $d - 1$ hidden layers with $n$ neurons each, there are $(n!)^{d-1}$ equivalent configurations corresponding to the permutation of neuron indices in each layer of the network (Goodfellow et al., 2016; Bishop, 1995). The permutation symmetries give rise to a loss landscape where any given global minimum in the weight space must have $(n!)^{d-1} - 1$ completely equivalent *partner* minima. This property of neural network landscapes is called *weight-space symmetry*.

Several (Saad and Solla, 1995; Amari et al., 2006; Wei et al., 2008) works explored the implications of weight-space symmetry for training dynamics in two-layer networks and found that training dynamics slow down near the singular regions caused by weight-space symmetry. Dauphin et al. (2014); Orhan and Pitkow (2017) argue that optimization paths may get close to the singular regions induced by weight-space symmetry and this, in turn, slows down training for deep neural networks. Exploiting weight-space symmetries, we give insights into and partial explanations of three observations on neural network landscapes.

**Observation 1.** Training dynamics are slow near singular regions caused by weight-space symmetry and stochastic gradient descent might travel near these regions throughout training (Saad and Solla, 1995; Wei et al., 2008; Amari et al., 2006; Dauphin et al., 2014; Orhan and Pitkow, 2017).

**Observation 2.** The Hessian of the loss function has numerous almost-zero eigenvalues throughout training, thus the landscape is flat in many directions (Sagun et al., 2017; Papyan, 2018; Ghorbani et al., 2019).

*Related to observation 1 and 2, we prove the existence of numerous connected high-dimensional plateaus extending across the landscape due to weight-space symmetries.*

**Observation 3.** The number of saddles can grow exponentially in neural network landscapes (Auer et al., 1996; Dauphin et al., 2014; Choromanska et al., 2015).

*Related to observation 3, we prove that there are at least polynomially many more saddles than the global minima due to weight-space symmetries in neural networks, without any further assumptions.*

In addition, we propose a novel low-loss path finding algorithm to find barriers between partner minima. We start from the known permutation symmetries and consider continuous low-loss paths that connect two equivalent global minima by merging the weight vectors of two neurons in a specific way. At a so-called *permutation point*, where the distance between the input and output weight vectors of the two neurons vanishes, the indices of the two neurons can be interchanged at no extra cost. After the change, the system returns on the 'mirrored' path back to the original configuration — except for the permutation of one pair of indices.

Surprisingly, we find that we can permute *all* neuron indices in the same layer at the same cost as the loss at a permutation point reached by moving along the path that merges a *single* pair of neurons. These constant-loss permutations are possible because each permutation point lies in a high-dimensional plateau of critical points. Our theory can be extended to higher-order saddles and provides explicit lower bounds for the number of first- and higher-order permutation points. Numerically, we confirm the existence of first-order permutation saddles.

In particular, the specific contributions of our work are:

- A simple low-loss path-finding algorithm linking partner global minima via a permutation point, implemented by minimization under a single scalar constraint (distance of weight vectors).
- The theoretical characterization of permutation points, for example that these are critical points and several permutation points are connected via paths at equal loss.
- A lower bound for the number of first- and higher-order permutation points and their corresponding plateaus.
- Numerical demonstrations of the path finding method in multilayer neural networks trained on MNIST.

## 1.1 RELATED WORK

**Structure of the landscape.** For linear networks, it was shown that all the critical points – except for the global minimum – are saddles in the case of two-layer (Baldi and Hornik, 1989) or multilayer networks (Freeman and Bruna, 2016; Kawaguchi, 2016; Lu and Kawaguchi, 2017). Interestingly, deep linear networks are reported to exhibit sharp transitions at the edges of extended plateaus (Saxe et al., 2013), similar to the plateaus observed in deep nonlinear networks (Goodfellow et al., 2014). For nonlinear multilayer networks, Choromanska et al. (2015) argue that all local minima lie below a certain loss value by drawing connections to the spherical spin-glass model. Improving upon this result, Soudry and Carmon (2016); Nguyen and Hein (2017) prove that almost all local minima are global minima for multilayer networks under mild over-parametrization assumptions.
**Bottom of the landscape.** Another line of research studies the bottom of the landscape containing global minima and low-loss barriers between them. Freeman and Bruna (2016) prove the existence of low-loss paths connecting global minima for wide two-layer networks by upper-bounding the loss along the path with a parameter that depends on the number of parameters and data smoothness. Draxler et al. (2018) use Nudged Elastic Band method introduced in Jónsson et al. (1998) to connect independent minima and numerically find that the barrier vanishes consistently for increasing

width and depth in DenseNet, ConvNet and ResNet architectures trained on CIFAR datasets. In a simultaneous work, Garipov et al. (2018) confirm that there is no significant barrier by connecting independent minima with polygonal chains.

**Training dynamics in the landscape.** For general loss functions, Lee et al. (2016) show that gradient descent with sufficiently small step-size converges to local minima if all the saddles have at least one negative eigenvalue. For overparametrized neural networks, gradient descent converges to global minima without moving far from initialization (Jacot et al., 2018; Du et al., 2018a;b), thus suggesting convex-like behavior around random initialization. For the finite size networks, how training dynamics converge to a minima and in particular *how fast* they converge remain an open question. For soft-committee machines, it turns out that the initial learning dynamics are slowed down by correlation of hidden neurons (Saad and Solla, 1995; Engel and Van den Broeck, 2001; Inoue et al., 2003). Amari et al. (2006); Wei et al. (2008) show that training dynamics slow down near singular regions due to weight-space symmetry. Dauphin et al. (2014) empirically argue that the large number of saddle points in the landscape makes training slow. Orhan and Pitkow (2017) numerically find that stochastic gradient descent may slow down near plateaus due to weight-space symmetry for deep (30 layers) feedforward networks trained on CIFAR100.

In this paper we show that there is an impressively large number of permutation points. Each permutation point is a critical point (either a local minimum or a saddle) with a large number of flat directions, potentially linked to the empirically observed plateaus. In contrast to an earlier study by Fukumizu and Amari (2000) with a scalar output for two-layered networks where a line of critical points around the permutation point was reported, we study a deep network with $d-1$ hidden layers and find multi-dimensional equal-loss plateaus. Moreover, we give a novel lower bound on the number of permutation points and construct sample paths between global minima using an algorithm that is different from previously used methods (Garipov et al., 2018; Draxler et al., 2018), since it exploits the symmetries at the permutation point.

## 1.2 PRELIMINARIES

We study multilayer neural networks $f(\boldsymbol{x}; \boldsymbol{\theta})$ with input $\boldsymbol{x} \in \mathbb{R}^{n_0}$, $d$ layers of $n_1, \ldots, n_d$ neurons per layer, parameters $\boldsymbol{\theta} = \{ \boldsymbol{W}^{(k)} \in \mathbb{R}^{n_k \times n_{k-1}} \text{ and } \boldsymbol{b}^{(k)} \in \mathbb{R}^{n_k} : k \in \{1, \ldots, d\} \} \in \Theta$ and $n_d$-dimensional output

$$f(\boldsymbol{x}; \boldsymbol{\theta}) = \boldsymbol{W}^{(d)} g \bigg( \cdots g \Big( \boldsymbol{W}^{(2)} g \big( \boldsymbol{W}^{(1)} \boldsymbol{x} + \boldsymbol{b}^{(1)} \big) + \boldsymbol{b}^{(2)} \Big) \cdots \bigg) + \boldsymbol{b}^{(d)} , \qquad (1)$$

where $g$ is a nonlinear activation function that operates component-wise on any vector.

**Definition 1 & 2. (Parameter vector)** We define the *parameter vector* $\boldsymbol{\vartheta}_m^{(k)}$ of neuron $m$ in layer $k$ as the incoming weights to a neuron $m$ in layer $k$ concatenated with its bias term: $\boldsymbol{\vartheta}_m^{(k)} = \big[ \boldsymbol{W}_{m,1}^{(k)}, \ldots, \boldsymbol{W}_{m,n_{k-1}}^{(k)}, \boldsymbol{b}_m^{(k)} \big]$. **(Output weight vector)** We define the *output weight vector* of neuron $m$ in layer $k$ as its outgoing weights from neuron $m$ in layer $k$ to the next layer: $\big[ \boldsymbol{W}_{1,m}^{(k+1)}, \ldots, \boldsymbol{W}_{n_{k+1},m}^{(k+1)} \big]$.

Since one can permute the neurons within each layer without changing the network function $f(\boldsymbol{x}; \boldsymbol{\theta})$, any point $\boldsymbol{\theta}$ induces a 'permutation set'. **Definition 3. (Permutation set)**

$$P(\boldsymbol{\theta}) = \big\{ \boldsymbol{\theta}' \in \Theta : \boldsymbol{W}'^{(k)}_{\sigma^{(k)}(i), \sigma^{(k-1)}(j)} = \boldsymbol{W}^{(k)}_{i,j} \text{ and } \boldsymbol{b}'^{(k)}_{\sigma^{(k)}(i)} = \boldsymbol{b}^{(k)}_i, k \in \{1, \ldots, d\} \big\}, \qquad (2)$$

of points $\boldsymbol{\theta}'$ with $f(\boldsymbol{x}; \boldsymbol{\theta}) = f(\boldsymbol{x}; \boldsymbol{\theta}')$, where $\sigma^{(k)}$ are permutations of the neuron indices $\{1, \ldots, n_k\}$ in (hidden) layer $k$ where $\sigma^{(0)}$ and $\sigma^{(d)}$ are fixed trivial permutations, since we want to permute neither the indices of the input nor that of the output. We will use the notation $\boldsymbol{\theta}' = \sigma_{l \Leftrightarrow m}^{(k)}(\boldsymbol{\theta})$ to indicate a point $\boldsymbol{\theta}'$ that differs from $\boldsymbol{\theta}$ only by swapping neurons $l$ and $m$ in layer $k$.

Note that the cardinality of a permutation set is maximal with $|P(\boldsymbol{\theta})| = \prod_{j=1}^{d-1} n_j!$ only if all parameter vectors $\boldsymbol{\vartheta}_l^{(k)} \neq \boldsymbol{\vartheta}_m^{(k)}$ are distinct for every $l \neq m$ and layer $k \in \{1, \ldots, d-1\}$. In the following, we will assume that, at global minima, all parameter vectors are distinct at every layer $k$.

**Definition 4. (Permutation point)** Consider a minimum of a multilayer network with $(n_1, \ldots, n_k - 1, \ldots, n_d)$ neurons per layer. We can map this minimum to a configuration in the landscape of a

multilayer network with $(n_1, \ldots, n_k, \ldots, n_d)$ neurons per layer by duplicating one neuron $m \in \{1, \ldots, n_k - 1\}$ in layer $k$ as follows: (i) substitute the parameter vector of the new neuron with a copy of the parameter vector $\boldsymbol{\vartheta}_m^{(k)}$ of neuron $m$, (ii) replace the output weight vector of the new neuron and the duplicated neuron $m$ with the initial output weight of neuron $m$ rescaled by $\frac{1}{2}$, and (iii) keep all the other parameters the same. This new configuration where the parameter vectors and the output weight vectors of the new neuron and the duplicated neuron $m$ are the same will be called a *permutation point*, denoted by $\boldsymbol{\theta}_{l \Leftrightarrow m}^{(k)}$, i.e. $\boldsymbol{\theta}_{l \Leftrightarrow m}^{(k)} = \sigma_{l \Leftrightarrow m}^{(k)}(\boldsymbol{\theta}_{l \Leftrightarrow m}^{(k)})$.

For training data $D = \{(\boldsymbol{x}^\mu, y^\mu) : \mu \in \{1, \ldots, T\}\}$ with targets $y^\mu \in \mathcal{Y}$, we define a loss function $L(\boldsymbol{\theta}; D) = \frac{1}{T} \sum_{\mu=1}^{T} \ell(y^\mu, f(\boldsymbol{x}^\mu; \boldsymbol{\theta}))$, where $\ell : \mathcal{Y} \times \mathbb{R}^{n_d} \to \mathbb{R}$ is some single-sample loss function. To simplify notation we will usually omit the explicit mentioning of the data in the loss function, i.e. $L(\boldsymbol{\theta}) \equiv L(\boldsymbol{\theta}; D)$.

A                                                     B

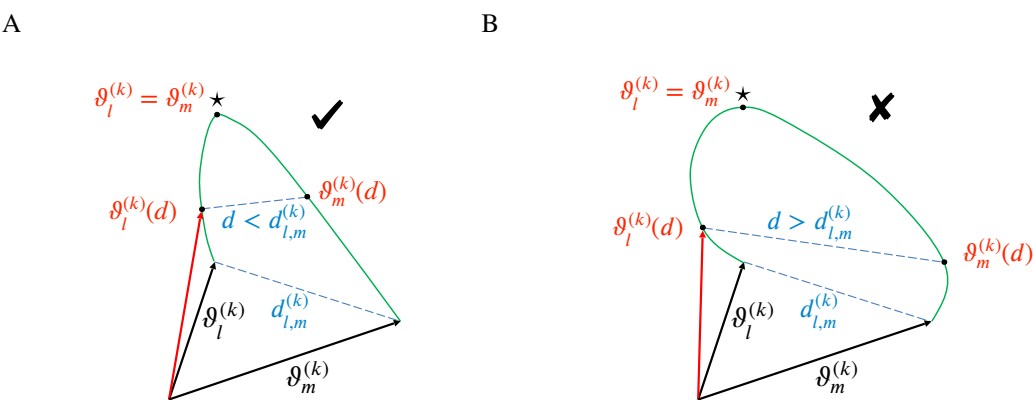

Figure 1: **A**. Configuration of two parameter vectors $\boldsymbol{\vartheta}_l^{(k)}$ and $\boldsymbol{\vartheta}_m^{(k)}$ (black) at a minimum $\boldsymbol{\theta}$ and a potential path (green) towards a permutation point $\boldsymbol{\theta}_{l \Leftrightarrow m}^{(k)}$ ($\star$). The path is parametrized by the distance $d$. Along the path the distance $d$ (blue dashed lines) decreases continuously starting at $d_{l,m}^{(k)}(\boldsymbol{\theta})$. Note that the path can lead to a permutation point far away from the initial configuration, outside the linear subspace spanned by the parameter vectors $\boldsymbol{\vartheta}_l^{(k)}$ and $\boldsymbol{\vartheta}_m^{(k)}$. **B**. We exclude hypothetical paths where the distance along the path increases.

## 2 MAIN RESULTS

In this section, we will first present a novel method to find a low-loss path between partner minima. Our method ensures that this path passes through a 'permutation point'. We study the properties of permutation points. Furthermore, we will introduce higher-order permutation points and provide a lower-bound on their number.

### 2.1 A NOVEL METHOD TO CONSTRUCT LOW-LOSS PATHS BETWEEN PARTNER MINIMA

One natural question regarding the geometry of the bottom of the landscape is the following: is it possible to find a continous low-loss path that connects two minima? In this work, we are interested in finding the barriers between *partner* global minima. In particular, we want to find a continuous low-loss path $\boldsymbol{\gamma} : [0, 1] \to \Theta$ connecting two partner minima by first merging two parameter vectors and output weight vectors ('permutation point') and then completing the path using symmetry. We will first introduce some concepts to introduce this low-loss path between partner minima formally.

**Definition 5. (Distance function)** $d_{l,m}^{(k)} : \Theta \to \mathbb{R}^+$ is a *distance function* that takes a configuration $\boldsymbol{\theta}$ and returns the squared Euclidean distance between the parameter vectors of neuron $l$ and $m$ at layer $k$:

$$d_{l,m}^{(k)}(\boldsymbol{\theta}) = \|\boldsymbol{\vartheta}_l^{(k)} - \boldsymbol{\vartheta}_m^{(k)}\|^2 \tag{3}$$

Our idea is to find a low-barrier path $\gamma_* = \arg\min_{\gamma:[0,\frac{1}{4}]\to\Theta} \max_{t\in[0,\frac{1}{4}]} L(\gamma(t))$ under the following constraints for the initial ($t = 0$) and quarter-way ($t = \frac{1}{4}$) configurations: $\gamma_*(0) = \theta$, where $\theta$ is the parameter configuration at the minimum, and $d^{(k)}_{l,m}(\gamma_*(\frac{1}{4})) = 0$. Furthermore, the distance between parameter vectors $\vartheta^{(k)}_l$ and $\vartheta^{(k)}_m$ in layer $k$ is decreasing, i.e. $d^{(k)}_{l,m}(\gamma_*(t)) < d^{(k)}_{l,m}(\gamma_*(t'))$ for all $t > t' \in [0, \frac{1}{4}]$[1] (see Fig. 1 and pseudocode in Appendix).

The quarter-way configuration guarantees that the parameter vectors are identical $\vartheta^{(k)}_l = \vartheta^{(k)}_m$, but puts no constraints on the output weights of neurons $m$ and $l$ in layer $k$. We can continously move from the quarter-way configuration ($t = \frac{1}{4}$) to a configuration at $t = \frac{1}{2}$ where the outputs weights of the related neurons are equal, without making any changes to the network output or the loss $L$ as follows: we will increase all output weights $W^{k+1}_{n,l}$ of neuron $l$ and decrease the corresponding output weights $W^{k+1}_{n,m}$ of neuron $m$ by the same amount continuously so as to keep their sum fixed until $W^{k+1}_{n,l} = W^{k+1}_{n,m}$ for every neuron $n \in \{1, \ldots, n_{k+1}\}$ at layer $k + 1$ (see Appendix Fig. 4).

**Lemma 1.** The configuration at $t = \frac{1}{2}$ is one of the permutation points.

Once we have reached $t = \frac{1}{2}$, we interchange the neuron indices of the 'merged' neurons and continue on the 'mirror' path that results from walking the first half of the path backwards with interchanged neuron indices, until we arrive at the partner minimum at $t = 1$, i.e. $\gamma(\frac{1}{2} + \tau) = \sigma^{(k)}_{l\Leftrightarrow m}(\gamma(\frac{1}{2} - \tau))$ for $\tau \in [0, \frac{1}{2}]$.

To find such paths algorithmically, we reparametrize $\vartheta^{(k)}_m(t) = \vartheta^{(k)}_l(t) + d(t)e(t)$ where $d(t)$ is a positive scalar and $e(t)$ is a unit-length vector. We start with $d(0) = d^{(k)}_{l,m}(\theta)$ and initialize $e$ in direction of the difference $\vartheta^{(k)}_m - \vartheta^{(k)}_l$ at the global minimum i.e., the initial parameter configuration. Next, we decrease $d$ infinitesimally and perform gradient descent for fixed $d$ on the loss $L$ until convergence. Note that all parameters can change, including $\vartheta^{(k)}_l$ and $e$ during gradient descent. This procedure is repeated until $d = 0$ at $t = \frac{1}{4}$. Finally we shift the respective output weights to the same value without changing the network function (see Appendix Fig. 4).

Since the path connects two partner minima, there must be at least one saddle point on the path $\gamma(t), t \in [0, 1]$, potentially but not necessarily, at the permutation point. Moreover, there is no guarantee that the highest saddle should be located at the permutation point (see Appendix Fig. 5).

## 2.2 CHARACTERIZATION OF PERMUTATION POINTS

In an earlier work, Fukumizu and Amari (2000) studied a specific set of critical points induced by the hierarchical structure in the neural network landscapes in two-layer neural networks. Let $L^{(H)}$ be the loss function ('landscape') of a two-layer neural network with $H$ neurons in the hidden layer and a single output. They showed that any critical point in the landscape of $L^{(H-1)}$ induces a line of critical points in the landscape of $L^{(H)}$. We study permutation points in the general setup for the neural networks with multiple outputs and multiple layers.

**Theorem 1**. (Fukumizu and Amari, 2000) By duplicating one parameter vector of any critical point in $L^{(H-1)}$ and keeping the sum of the two output weights corresponding to the duplicated parameter vectors fixed at the value of the original output weight, one obtains a line of critical points in $L^{(H)}$. [2]

**Proposition 1**. (i) Permutation points $\theta^{(k)}_{l\Leftrightarrow m}$ are critical points of the original loss function.

(ii) Any permutation point lies inside a $n_{k+1}$-dimensional equal-loss subspace of critical points.

(iii) All other permutations of neuron indices in layer $k$ can be performed by continuous equal-loss transformations starting from permutation points $\theta^{(k)}_{l\Leftrightarrow m}$ of neurons $l$ and $m$, i.e. there is a continuous

---

[1]Note that the parametrization $t$ including the exact $t$ values for the mentioned configurations are arbitrary and only used for conceptualizing the path. The relevant parametrization of the path will be the distance $d$.

[2]Theorem 1 is easily verified by setting the derivatives of all parameters to zero in $L^{(H-1)}$ (critical point) and observing that the derivatives of the corresponding configuration in $L^{(H)}$ will be zero under the mapping stated in the theorem.

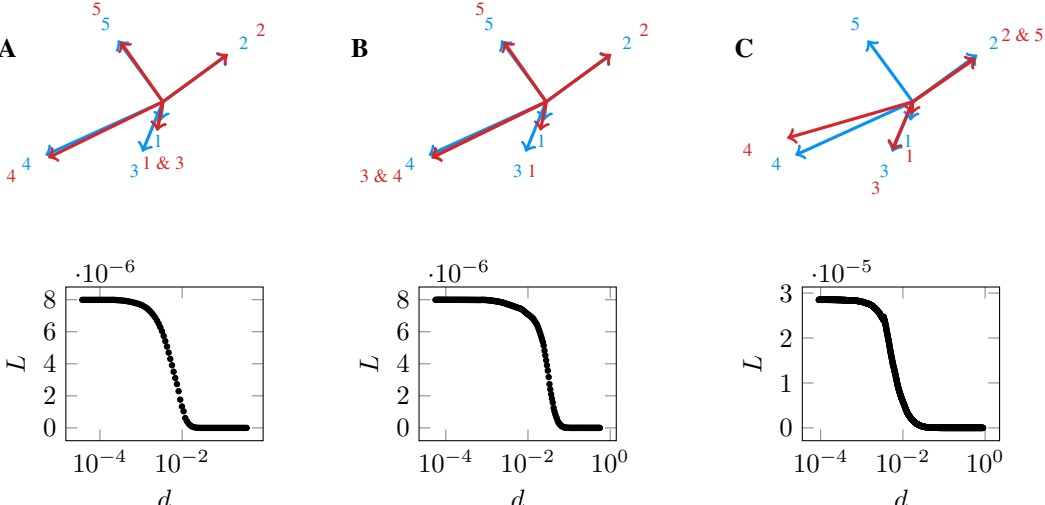

Figure 2: **Paths to the same or to different permutation points. Top row**. Configuration of the 5 weight vectors $\boldsymbol{W}_{i,:}^{(1)}/\boldsymbol{b}_i^{(1)}$ of the first layer at the global minimum (blue) and at a permutation point (red) reached after merging the parameter vectors of two neurons. Note that the global minimum (i.e., the starting configuration) is the same for (A), (B) and (C) but the loss at the permutation point can be the same – (A) and (B) – or different – (A) and (C) – depending on the pair of neurons chosen for merging. Numbers indicate neurons. **Bottom row**. Quadratic loss $L$ as a function of the distance $d$ between the neurons to be merged. The distance was decreased in 200 logarithmically spaced steps from $d = d_{l,m}^{(1)}(\boldsymbol{\theta}^*)$ to $1/10^4$ of the initial value. For each $d$, full batch gradient descent on the loss $L$ was performed until convergence. Training data was generated by sampling $10^3$ two-dimensional input points $\boldsymbol{x}^\mu$ from a standard normal distribution and computing labels $y^\mu = f(\boldsymbol{x}^\mu; \boldsymbol{\theta}^*)$ using a teacher network (shown in blue, equivalent to the configuration at the global minima). The teacher had a single layer of five hidden neurons with rectified-linear activation function $g$ and one linear output layer.

path $\boldsymbol{\rho} : [0,1] \to \Theta$ such that $\boldsymbol{\rho}(0) = \boldsymbol{\theta}_{l \Leftrightarrow m}^{(k)}$ and $\boldsymbol{\rho}(1) = \boldsymbol{\theta}_{i \Leftrightarrow j}^{(k)}$ and $L(\boldsymbol{\rho}(t)) = L(\boldsymbol{\rho}(0))$ for all $t \in [0,1]$ and all $i \neq j \in \{1, \ldots, n_k\}$. (Proofs: see appendix).

Therefore, each of the $\frac{n_k(n_k-1)}{2}$ different permutation points of layer $k$ corresponds to a plateau of $n_{k+1}$ dimensions. This plateau enables the exchange of all indices in layer $k$. Note that there can be multiple plateaus on different loss levels that correspond to different local minima of the smaller networks where one of the neurons is dropped at a permutation point. For example in Fig. 2 one can exchange all indices in the hidden layer through the configuration in A and B or through the configuration in C that has another loss level. Note that, amongst all these permutation points embedded in different plateaus, we could for example search for the one with the lowest cost —and this lowest-cost permutation would then also connect all global minima caused by arbitrary permutations of neurons in layer $k$.

**Definition 6. (Higher-order permutation point)** Consider a minimum of a multilayer network with $(n_1, \ldots, n_k - K, \ldots, n_d)$ neurons per layer. We can map this minimum to a configuration in the landscape of a multilayer network with $(n_1, \ldots, n_k, \ldots, n_d)$ neurons per layer by replicating some neurons $m_j \in \{1, \ldots, n_k - K\}$ in layer $k$ to fill out the parameters of new neurons as follows: (i) substitute the parameter vectors of the new neurons with one of the parameter vectors of the initial minimum, (ii) replace the output weight vectors of the new neurons and the output weight of the corresponding parameter vector $m_j$ with the initial output weight of the replicated neuron $m_j$ normalized so that the mentioned output weight vectors sum up to the original output weight vectors, and (iii) keep all the other parameters the same. This new configuration where the parameter vectors and the output weight vectors of the new neurons and the replicated neuron $m_j$ for several $j$ in layer $k$ are the same will be called a $K$-th order permutation point.

This natural generalization on the 1-st order permutation points to higher-orders enables generalizing Proposition 1(i) and (ii) to the $K$-th order permutation points.

**Proposition 2.** (i) A $K$-th order permutation point is a critical point of the original landscape. (ii) Any $K$-th order permutation point at layer $k$ lies in a $Kn_{k+1}$-dimensional subspace of equal loss parameter configurations. (Proofs: see appendix).

### 2.3 Counting Higher-Order Permutation Points

A configuration in the landscape of $L^{(H-K)}$ can be mapped to an equivalent configuration in $L^{(H)}$ with the procedure described in Definition 6. We can then *count* the number of permutation points that reduce to the same configuration in $L^{(H-K)}$ combinatorially (see Appendix Fig. 6 for the explanation of combinatorial counting). For counting, we consider the cardinality of the permutation set of a permutation point. Since some parameter vectors are replicated, we have to consider permutations of sometimes identical neurons. This enables finding a lower bound on the number of critical points that have higher loss values than the global minima in general.

**Proposition 3.** In a neural network with $(n_1, \ldots, n_d)$ neurons per layer, let $T(K, n_k)$ denote the ratio of the number of $K^{\text{th}}$-order permutations points at layer $k$ to the number of global minima for $k = 1, \ldots, d-1$ and $K \geq 1$.

(i) For $K = 1, 2, 3$ and $n_k \geq 2K$, we find $T(K, n_k)$ to be:

- $T(K = 1, n_k) = \binom{n_k-1}{1} \frac{1}{2}$
- $T(K = 2, n_k) = \binom{n_k-2}{1} \frac{1}{3!} + \binom{n_k-2}{2} \frac{1}{2^2}$
- $T(K = 3, n_k) = \binom{n_k-3}{1} \frac{1}{4!} + \binom{n_k-3}{2} \frac{1}{3!} + \binom{n_k-3}{3} \frac{1}{2^3}$

(ii) For general $K \leq n_k/2$, we find the bound $T(K, n_k) \geq \binom{n_k-K}{K} \frac{1}{2^K}$. (Proofs: see appendix)

Considering all the layers, we note that the number of permutation points of order $K$ is at least $\sum_{k=1}^{d-1} \frac{1}{2^K} \binom{n_k-K}{K}$ times more than the global minima for $2K \leq \min_k n_k$.

**Lemma 2.** For finite $K \in \mathbb{Z}^+$ and $n_k \to \infty$, $T(K, n_k) > c_K n_k^K$, since $\frac{1}{2^K} \binom{n_k-K}{K} \to c_K n_k^K$.

When one layer has large number of neurons (i.e. $n_k \to \infty$) then the ratio $T(K, n_k)$ grows with $n_k^K$.

Every permutation point lies inside a high-dimensional subspace of equal loss (Proposition 2(ii), see Appendix Fig. 7 for illustration). Importantly, every permutation point lies inside a *distinct* but connected subspace. Therefore the count for permutation points holds for the corresponding high-dimensional equal-loss subspaces of critical points.

**Lemma 3.** In a neural network with $(n_1, \ldots, n_d)$ neurons per layer, there are (at least) $\sum_{k=1}^{d-1} T(K, n_k) \prod_{k=1}^{d-1} n_k!$ many $Kn_{k+1}$-dimensional equal-loss subspaces of critical points at the loss of a $K$-th order permutation point for $2K \leq \min_k n_k$.

We could start at an arbitrary configuration in consider the landscape of $L^{(H-K)}$ and the corresponding equal-loss high-dimensional subspaces in $L^{(H)}$, where each configuration in the subspace computes the same function as the initial configuration. This procedure again would yield the same number of high-dimensional equal-loss subspaces. Therefore due to weight-space symmetry, neural network landscapes do not only exhibit numerous high-dimensional plateaus of critical points but also numerous high-dimensional plateaus (of usually non-critical points) at various loss values.

## 3 Empirical results

Using a similar procedure as in the toy example (see Fig. 2), we constructed paths between global minima in a fully connected three-layer network with $n_1 = n_2 = H$ and $n_3 = 10$ neurons (see Fig. 3). In order to study *global* minima we used a student-teacher setting[3]: the teacher network was

---

[3]Paths between *local* minima could be constructed by starting with a pre-trained network on MNIST and applying our path finding method with standard gradient descent training on MNIST. We expect to find similar results.

pre-trained on the MNIST data set using negative log-likelihood loss and its parameters $\theta^*$ were kept fixed thereafter. We initialized the student with the parameters $\theta^*$ of the teacher and decreased the distance $d$ between the parameter vectors of two selected neurons $m$ and $l$ in layer $k = 2$ in 100 logarithmically spaced steps from $d_{m,l}^{(2)}(\theta^*)$ to $1/10^4$ of the original value. For every value of $d$, the student was trained on a regression task with a mean-squared error loss $L$ between teacher and student output using full batch gradient descent until convergence. With $y^\mu = f(\boldsymbol{x}^\mu; \theta^*) \in \mathbb{R}^{10}$ being the output of the last layer before the softmax operation, we chose $L = \frac{1}{T} \sum_{\mu=1}^{T} \|y^\mu - f(\boldsymbol{x}^\mu; \theta)\|^2 / \langle y_i^{\mu 2} \rangle$ as the mean squared error loss between teacher and student, where $\langle . \rangle$ denotes the mean over patterns and dimensions and $\mu = 1, \dots, T$ enumerates the samples of the data set. Apart from a few cases, where the trajectory towards the permutation point passed through a saddle on the way, in most cases the loss increased monotonically until the permutation point. This indicates that the permutation point is a saddle, and not a minimum. As expected from theoretical results (Freeman and Bruna, 2016) and empirically observed by (Draxler et al., 2018), the barrier height (loss at saddle) decreased with the number $H$ of hidden neurons per layer.

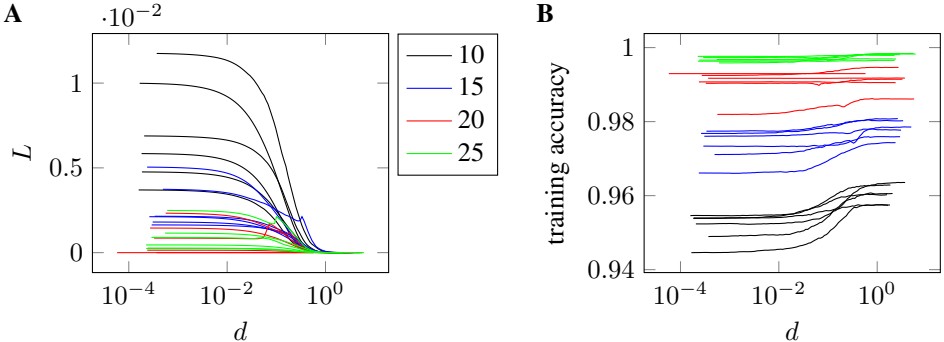

Figure 3: **A low-loss permutation path in the loss landscape of a multi-layer network using a student-teacher setup trained on MNIST**. We merged the parameter vectors of two neurons with high cosine-similarity in the second hidden layer of a three-layer student network. The corresponding teacher network with $H = 10, 15, 20$ or $25$ was trained on MNIST. For each hidden layer size we trained 6 teacher networks with different random seeds and display one curve per hidden layer size and seed. **A.** In most cases, the mean squared loss $L$ between teacher and student output increases monotonically along our constructed paths from a global minimum until the permutation point. In these cases the latter corresponds to the loss barrier along the path. Note that the barrier height (loss at saddle) decreases with $H$. **B.** The MNIST classification accuracy on the training set decreases only marginally when moving to a permutation point.

## 4 DISCUSSION

The surprising training performance of neural networks despite their highly non-convex nature has been drawing attention to the structure of the loss landscape. In this paper, we explored how weight-space symmetry induces saddles and plateaus in the neural network loss landscape. We found that special critical points, so-called *permutation points*, are embedded in high-dimensional flat plateaus. We proved that all permutation points in a given layer are connected with equal-loss paths, suggesting new perspectives on loss landscape topology. We provided a novel lower bound for the number of first- and higher-order permutation points and proposed a low-loss path finding method to connect equivalent minima. The empirical validation of our path finding algorithm in a multilayer network trained on MNIST showed that permutation points could indeed be reached in practice. Additionally, we observed that the loss at the permutation point (barrier) decreased with network size and thus confirmed Freeman and Bruna (2016)'s findings for loss barriers between global minima. High-dimensional flat regions around permutation points could be one of the causes of the empirically observed slow phases in training.

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

# A Supplementary Figures

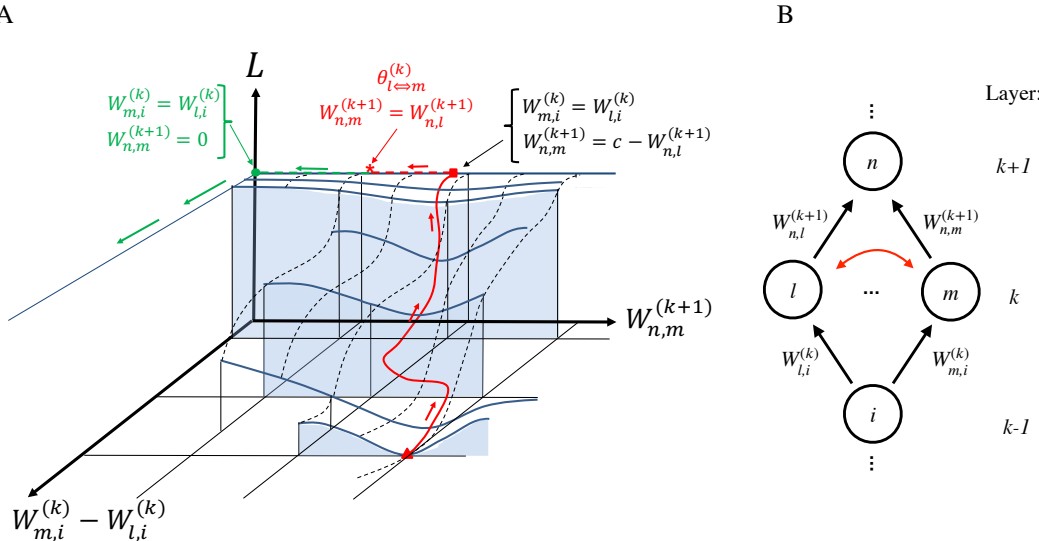

Figure 4: **A**. The loss landscape (schematic) as a function of two parameters: the difference $W_{m,i}^{(k)} - W_{l,i}^{(k)}$ between the weights from neuron $i$ to neurons $m$ and $l$ in layer $k$ and the weight $W_{n,m}^{(k+1)}$ from neuron $m$ to neuron $n$ in the next layer (see **B** for network graph). The red curve indicates the path from one of the global minima (red triangle) to a half-way configuration where the difference between the input weight vectors of neurons $m$ and $l$ in layer $k$ vanishes ($W_{m,i}^{(k)} = W_{l,i}^{(k)}$, red square). Along the axis $W_{m,i}^{(k)} - W_{l,i}^{(k)} = 0$, we can change the output weight $W_{n,m}^{(k+1)}$ at constant loss (dashed horizontal line), as long as the sum $W_{n,m}^{(k+1)} + W_{n,l}^{(k+1)} = c$ remains constant. The point $W_{n,m}^{(k+1)} = W_{n,l}^{(k+1)}$ where the two output weights are identical (and assuming the same matching condition for the other output weights) defines the permutation point $\boldsymbol{\theta}_{l \Leftrightarrow m}^{(k)}$ (red $\star$) where we can swap the indices of neurons $m$ and $l$ in layer $k$ at equal loss and continuously in parameter space. If we then shift (dashed green line) all output weights of neuron $m$ in layer $k$ to zero ($W_{n,m}^{(k+1)} = 0$ for all $n$, green filled circle), we are free to change the weight $W_{m,i}^{(k)}$ at constant loss (green arrows) so as to perform further permutations of neurons in layer $k$.

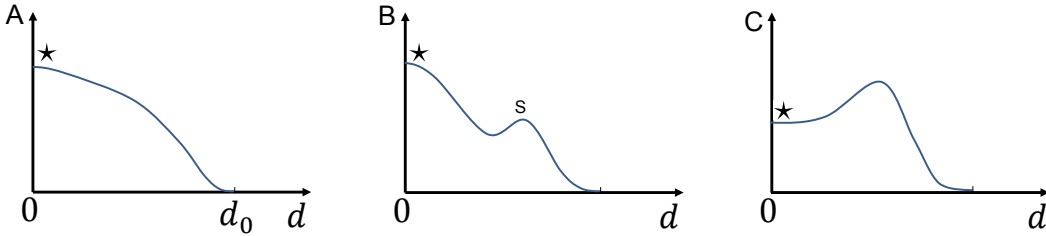

Figure 5: Loss $L$ (vertical axis) on the permutation path as a function of the distance $d$ between the two parameter vectors to be permuted (schematic). **A**. In the teacher network the two parameter vectors have a distance $d_0$. Along the path, the distance is reduced to zero. At the permutation point ($\star$), the loss reaches a maximum which corresponds to a saddle point of the total loss function. **B**. On the path towards the permutation point ($\star$) an intermediate saddle point (S) may occur. **C**. The permutation point ($\star$) could be a minimum along the path.

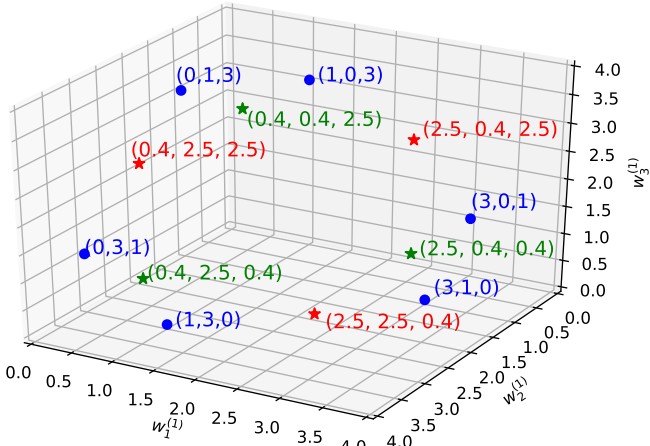

Figure 6: Visualizing how permutation points arise in between global minima for a (hypothetical) network function with scalar inputs and one hidden layer with three neurons. Here we do not consider biases for simplicity. Blue dots: 3! many equivalent global minima. A red $\star$: One permutation point (of first order at layer $k = 1$) represented by its weights in the first layer, i.e. $(2.5, 2.5, 0.4)$. All red $\star$: One permutation set where the weight value $2.5$ is duplicated. All green $\star$: The other permutation set where the weight value $0.4$ is duplicated. Note that overall, we have $6$ permutation points that give rise to the same network function as the weight configuration with two hidden neurons with $(2.5, 0.4)$. Indeed, $T(K = 1, n_1 = 3) = \binom{3-1}{1} \frac{1}{2} = 1$ confirms why the number permutation points corresponding to this particular weight configuration with two hidden neurons is equal to the number of global minima. Note that the weight values are assigned randomly for visualization purposes.

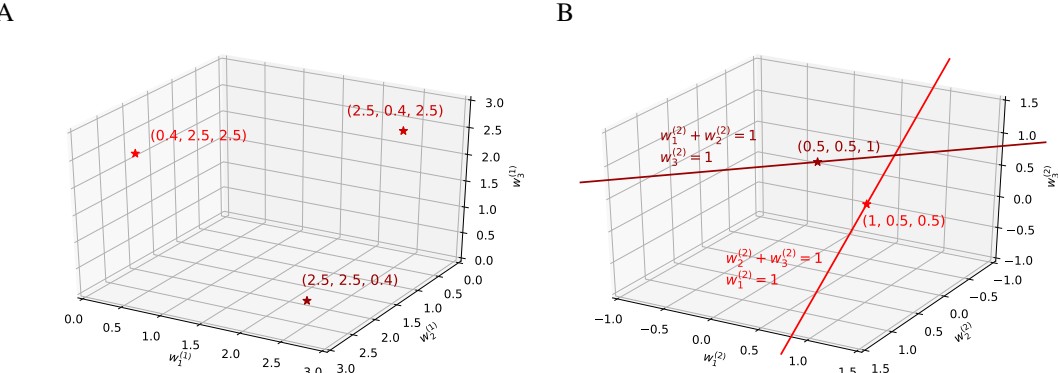

Figure 7: **A**. Zooming in permutations points of one of the permutation sets (red ⋆ in Fig. 6). **B**. Visualizing how permutation points (at layer $k = 1$, see **A**) lie inside equal-loss lines in the weight space of the layer $k + 1 = 2$. Only two out of three lines are shown for simplicity. We observe that the number of such equal-loss lines (hyperplanes) is equal to the number of permutation points, i.e. each permutation point lies inside one distinct line.

## B  PATH FINDING ALGORITHM

---

**Algorithm 1** Finding Low-Loss Paths through Permutation Points

---
**Require:** parameters $\boldsymbol{\theta}$, indices $k, l, m$, SCHEDULE of decreasing distances
1:  $d_0 \leftarrow d_{l,m}^{(k)}(\boldsymbol{\theta})$

2:  **Re-parametrize** $\boldsymbol{\theta} \rightarrow \tilde{\boldsymbol{\theta}} = (\{\tilde{\boldsymbol{\vartheta}}_i^{(j)}\}_{(i,j)\neq(m,k)}, \boldsymbol{e}, d)$ s.t. $\boldsymbol{\vartheta}_i^{(j)} = \tilde{\boldsymbol{\vartheta}}_i^{(j)}, \boldsymbol{\vartheta}_m^{(k)} = \tilde{\boldsymbol{\vartheta}}_l^{(k)} + d\boldsymbol{e}/\|\boldsymbol{e}\|$

3:  $d \leftarrow d_0, \boldsymbol{e} \leftarrow \boldsymbol{\vartheta}_m^{(k)} - \boldsymbol{\vartheta}_l^{(k)}$
4:  **for all** $d'$ in SCHEDULE$(d_0)$ **do**
5:      $d \leftarrow d'$
6:      $\gamma(d') \leftarrow \tilde{\boldsymbol{\theta}}(d')$ as found by gradient descent on $L(\tilde{\boldsymbol{\theta}} \setminus \{d\})$
7:  **end for**
8:  **return** $\gamma$

---

## C  PROOFS

### C.1  HALF-WAY CONFIGURATIONS ARE PERMUTATION POINTS

**Proof of Lemma 1.** Since the parameter vectors and output vectors of the two neurons are identical at $t = \frac{1}{2}$, we can merge the two neurons into a single one (by appropriately rescaling the corresponding output weights) so that the number of neurons $n_k$ in layer $k$ is reduced by one. Since our path-finding method minimizes the gradient, the configuration at $t = \frac{1}{2}$ is a local minimum of the smaller network, hence a permutation point.

Alternatively, we will reformalize the path-finding problem with Lagrange multipliers: minimize $L(\boldsymbol{\theta}) + \lambda d_{l,m}^{(k)}(\boldsymbol{\theta})$ for increasing values of $\lambda$, starting with $\lambda = 0$ and increasing (potentially) until $\lambda \rightarrow \infty$. The goal here is to decrease the distance down to zero while keeping the loss minimal. For every $\lambda$, we will obtain a minimizer $\boldsymbol{\theta}$. This sequence of $\boldsymbol{\theta}$ configurations will be an approximate discretization of the ideal path $\gamma_*$. With the Lagrangian formulation, we can easily show that the half-way configuration is a critical point: it is a minimizer of $L(\boldsymbol{\theta}) + \lambda d_{l,m}^{(k)}(\boldsymbol{\theta})$ when $d_{l,m}^{(k)}(\boldsymbol{\theta}) = 0$ for a big enough $\lambda$. Therefore, $\nabla L(\boldsymbol{\theta}) + \lambda \nabla d_{l,m}^{(k)}(\boldsymbol{\theta}) = 0$. Note that the partial derivatives of $d_{l,m}$ with respect to the parameter vectors of neuron $l$ and $m$ in layer $k$ are $\frac{\partial d_{l,m}^{(k)}}{\partial \boldsymbol{\vartheta}_l^{(k)}} = 2(\boldsymbol{\vartheta}_l^{(k)} - \boldsymbol{\vartheta}_m^{(k)})$, $\frac{\partial d_{l,m}^{(k)}}{\partial \boldsymbol{\vartheta}_l^{(k)}} = 2(\boldsymbol{\vartheta}_m^{(k)} - \boldsymbol{\vartheta}_l^{(k)})$ and both are equal to zero at the half way configuration since $\|\boldsymbol{\vartheta}_l^{(k)} - \boldsymbol{\vartheta}_m^{(k)}\|^2 = 0$. All other partial derivatives with respect to other parameter vectors are $0$ since $d_{l,m}^{(k)}$ does not depend on them. Therefore $\nabla d_{l,m}^{(k)}(\boldsymbol{\theta}) = 0$. Consequently, $\nabla L(\boldsymbol{\theta}) = 0$. We note that the Lagrangian formulation is an approximation on the ideal path $\gamma_*$.

### C.2  PROPERTIES OF (FIRST-ORDER) PERMUTATION POINTS

**Proof of Proposition 1.** (i) Let us assume we merged the neurons $l$ and $m$ in layer $k$ and scaled the output weight vectors of these neurons with $\frac{1}{2}$. In that case, the output vector in layer $k + 1$ remains the same. The derivative with respect to an output in the layer $k + 1$ will be the same, i.e. $\frac{\partial L}{\partial a_n^k}$ for $n \in [n_{k+1}]$ will not change under this mapping. The derivatives with respect to the output weights (say $\boldsymbol{w}_i^k$ for $i \in [n_k]$) of layer $k$ is $\sum_{n=1}^{n_{k+1}} \frac{\partial L}{\partial a_n^{k+1}} \frac{\partial a_n^{k+1}}{\partial \boldsymbol{w}_i^k}$. These derivatives would be the same as the corresponding derivatives before the mapping is performed. The derivatives with respect to the parameter vectors of layer $k$ is $\frac{\partial L}{\partial a_m^k} \frac{\partial a_m^k}{\partial \boldsymbol{\vartheta}_m^k}$ remains the same up to a scaling in the output weight value.

All the other derivatives remain the same (up to a scalar constant) as the corresponding derivatives before the mapping is performed. Therefore, a critical point in the landscape of the neural network with $(n_1, \ldots, n_k - 1, \ldots, n_d)$ neurons per layer will map to another critical point in the landscape of the neural network with $(n_1, \ldots, n_k, \ldots, n_d)$ neurons per layer under the function preserving mapping described in the definition of a permutation point. This proposition is a straightforward

extension to the Theorem 1 in (Fukumizu and Amari, 2000) for multiple output and multiple-layer neural networks.

(ii) Once the parameter vectors of neurons $m$ and $l$ in layer $k$ are identical, they implement the same function. Any change of an output weight $\boldsymbol{W}_{n,m}^{(k+1)}$ preserves the network function, and criticality (see Proposition 1-(i) above), as long as the output weight $\boldsymbol{W}_{n,l}^{(k+1)}$ is coadapted so as to keep the sum $\boldsymbol{W}_{n,m}^{(k+1)} + \boldsymbol{W}_{n,l}^{(k+1)}$ constant. The sum-constraint $\boldsymbol{W}_{n,m}^{(k+1)} + \boldsymbol{W}_{n,l}^{(k+1)} = c$ for each $n$ in layer $k+1$ defines an $n_{k+1}$-dimensional hyperplane of critical points. In particular, at each point in the hyperplane, we have $n_{k+1}$ directions of the Hessian with zero Eigenvalues if the activation function $g$ is twice-differentiable.

(iii) We make the following sequence of continuous transformations that are all possible at fixed loss. First, we decrease the output weights of neuron $m$ to zero while increasing those of $l$ by the same amount, keeping the sum of weights $\boldsymbol{W}_{j,m}^{(k+1)} + \boldsymbol{W}_{j,l}^{(k+1)}$ constant for each $j$ in layer $k+1$. Second, we change smoothly the input parameter vector of neuron $m$ to match those of an arbitrary other neuron $i$ in the same layer $k$. Third, we increase the output weights of neuron $m$ while decreasing those of neuron $i$ until all output weights of neuron $i$ are zero, keeping the sum of weights $\boldsymbol{W}_{j,m}^{(k+1)} + \boldsymbol{W}_{j,i}^{(k+1)}$ constant for each $j$ in layer $k+1$. Fourth, we reduce the input parameter vector of neuron $i$ to zero. Fifth, we increase the input parameter vector of neuron $i$ to match that of neuron $l$ at the permutation point. Finally, we equally share output weights between neurons $i$ and $l$ so that $i$ has the same weights as previously neuron $m$ at the permutation point.

Effectively, this procedure enables us to exchange an arbitrary neuron $i$ with neuron $m$, but the procedure can be repeated for further permutations. The permutations constructed in the proof of property (ii) start at permutation points where the parameter vectors of neurons $l$ and $m$ merge. Therefore the loss associated with all the permutations constructed in the proof is $L(\boldsymbol{\theta}_{l \Leftrightarrow m}^{(k)})$, the one of this permutation point. However, we could also begin with two other weight vectors $i$ and $j$ and construct the path leading to another $\boldsymbol{\theta}_{i \Leftrightarrow j}^{(k)}$, that has, in general, a different loss.

## C.3 PROPERTIES OF HIGHER-ORDER PERMUTATION POINTS

**Proof of Proposition 2**. (i) We take a minimum in the landscape of neural network with $(n_1, \ldots, n_k - K, \ldots, n_d)$ neurons per layer. Using Proposition 1-(i), we map to a critical point in the landscape of neural network with $(n_1, \ldots, n_k - K + 1, \ldots, n_d)$ neurons per layer where one parameter vector in layer $k$ is duplicated and the corresponding output weights are rescaled with $\frac{1}{2}$. Repeating the same mapping starting at the latter critical point, we map to another critical point in the landscape of neural network with $(n_1, \ldots, n_k - K + 2, \ldots, n_d)$ neurons per layer. Repeating this mapping for $K$ times, we end up a $K$-th order permutation point (up to changes in the output weights of the replicated neurons as long as the network function is preserved) and this is a critical point in the landscape of neural network with $(n_1, \ldots, n_k, \ldots, n_d)$ neurons per layer by induction.

(ii) We will denote the parameter vectors of a neural network with $(n_1, \ldots, l = n_k - K, \ldots, n_d)$ neurons per layer with $\{\boldsymbol{\vartheta}_1^{(k)}, \boldsymbol{\vartheta}_2^{(k)}, \ldots, \boldsymbol{\vartheta}_l^{(k)}\}$ and that of $f(n_1, \ldots, n_k, \ldots, n_d)$ neurons with $\{\boldsymbol{\vartheta}_1'^{(k)}, \boldsymbol{\vartheta}_2'^{(k)}, \ldots, \boldsymbol{\vartheta}_{n_k}'^{(k)}\}$. Let's consider an unordered partition of $n_k = s_1 + s_2 + \ldots + s_l$ with $s_m \geq 1$ for $m = 1, \ldots, l$. Without loss of generality, let's assume that the first $s_1$ parameter vectors of layer $k$ are the same, then the next $s_2$ and so on. Equivalently, for all $m = 1, \ldots, l$, we have $\boldsymbol{\vartheta}_j'^{(k)} = \boldsymbol{\vartheta}_m^{(k)}$ for $j = s_{m-1} + 1, s_{m-1} + 2, \ldots, s_{m-1} + s_m$ where $s_0 = 0$.

The only free variables are the outgoing weights of the replicated neurons- except for the constraint that the summation of these weights is fixed. These constraints correspond to $\sum_{j=s_{m-1}+1}^{s_{m-1}+s_m} \boldsymbol{W}_{i,j}'^{(k+1)} = \boldsymbol{W}_{i,m}^{(k+1)}$ for all neurons $i$ at layer $k+1$. Overall, there are $(\sum_{m=1}^{l} s_m) n_{k+1} = n_k n_{k+1}$ free variables constrained by $l n_{k+1}$ equations. One equation defines a $n_k n_{k+1} - 1$ dimensional hyperplane in the $n_k n_{k+1}$ space. Intersecting $l n_{k+1}$ of these hyperplanes, we end up having a $(n_k - l) n_{k+1} = K n_{k+1}$ dimensional equal-loss hyperplane.

### C.4 COUNTING HIGHER-ORDER PERMUTATION POINTS

We simulate a smaller neural network with the big network of $(n_1, \ldots, n_k, \ldots, n_d)$ neurons across the $d$ layers. We assume that a minimum of the small neural network has $n_k - K$ distinct parameter vectors at layer $k$ and $n_j$ distinct parameter vectors at other layers $j \neq k$. A $K$-th order permutation point (at layer $k$) of the big network implements all the $(n_1, \ldots, n_k - K, \ldots, n_d)$ parameter vectors of the small network and only these. Since the big networks has $n_k$ neurons in layer $k$ and the small network only $n_k - K$, the big network must reuse some of these parameter vectors of the smaller network several times. Therefore we count the number of permutations of indices to calculate $T(K, n_k)$. We will start with $K = 1$.

**Proof of Proposition 3.**

**(1) The case $K = 1$:**

At a first-order permutation point in layer $k$, we have $l = n_k - 1$ distinct parameter vectors $\{\boldsymbol{\vartheta}_1^{(k)}, \boldsymbol{\vartheta}_2^{(k)}, \ldots, \boldsymbol{\vartheta}_l^{(k)}\}$ for a total of $n_k$ neurons. Therefore two of the $n_k$ neurons must have the same parameter vector. More formally, there is only one way to partition $n_k$ into $n_k - 1$ positive integers without respecting order and this unordered partition can be represented as $n_k = 2 + 1 + \ldots + 1$.

The duplicated parameter vector could be the first one, $\{\boldsymbol{\vartheta}_1^{(k)}\}$, or the second one, ... or the last one, $\{\boldsymbol{\vartheta}_l^{(k)}\}$. Therefore there are $\binom{n_k-1}{1}$ choices. For each of these choices (say, we double the third parameter vector), we have $\frac{n_k!}{2!}$ permutations of indices of neurons in layer $k$. If we include the permutations that are possible at all other layers, we have $\binom{n_k-1}{1} \frac{1}{2} \prod_{j=1}^{d-1} n_j!$ first-order permutation points (see Fig. 6). Since the cardinality of the permutation set induced by a global minimum is $|P(\boldsymbol{\theta})| = \prod_{j=1}^{d-1} n_j!$, we arrive at a ratio $T(K = 1, n_k) = \binom{n_k-1}{1} \frac{1}{2}$.

**(2) The case $K = 2$:**

There are two ways to have $n_k - 2$ distinct vectors out of $n_k$, corresponding to two unordered partitions of $n_k$: (i) $n_k = 3 + 1 + \ldots + 1$, and (ii) $n_k = 2 + 2 + 1 + \ldots + 1$.

(i) $n_k = 3 + 1 + \ldots + 1$
For this case, we have $\frac{n_k!}{3!}$ permutations given by permuting the neuron indices of layer $k$ instead of the usual $n_k!$ permutations since we should eliminate the equivalent permutations corresponding to the permutations among the replicated parameter vectors with a division by $3!$. Therefore, this 2-nd order permutation point induces a permutation set with cardinality $|P(\boldsymbol{\theta})| = \frac{1}{3!} \prod_{j=1}^{d-1} n_j!$.

Now we should consider other 2-nd order permutation points (at layer $k$) giving rise to the same network function and corresponding to the same unordered partition, which we did not count in this permutation set. If we had chosen another parameter vector to replicate three times, this 2-nd order permutation point would induce another permutation set. Note that we can choose the parameter vector to replicate out of $n_k - 2$ in $\binom{n_k-2}{1}$ ways and there are $\frac{1}{3!} \prod_{j=1}^{d-1} n_j!$ many points in each one of the permutation sets. Therefore, we end up having $\binom{n_k-2}{1} \frac{1}{3!} \prod_{j=1}^{d-1} n_j!$ many 2-nd order permutation points (at layer $k$) corresponding to this unordered partition.

(ii) $n_k = 2 + 2 + 1 + \ldots + 1$
For this case, we have $\frac{n_k!}{2!^2}$ permutations given by permuting the neuron indices of layer $k$ instead of the usual $n_k!$ permutations since we should eliminate the equivalent permutations corresponding to the permutations among the two pairs of duplicated parameter vectors with a division by $2!^2$. Therefore, this 2-nd order permutation point induces a permutation set with cardinality $|P(\boldsymbol{\theta})| = \frac{1}{2!^2} \prod_{j=1}^{d-1} n_j!$.

Again, we should consider other 2-nd order permutation points (at layer $k$) giving rise to the same network function and corresponding to the same unordered partition. Note that we can choose two parameter vectors to duplicate out of $n_k - 2$ in $\binom{n_k-2}{2}$ ways and there are $\frac{1}{2!^2} \prod_{j=1}^{d-1} n_j!$ many points in each one of the permutation sets. Therefore, we end up having $\binom{n_k-2}{2} \frac{1}{2!^2} \prod_{j=1}^{d-1} n_j!$ many 2-nd

order permutation points (at layer $k$) corresponding to this unordered partition. Overall, we have $\binom{n_k-2}{1}\frac{1}{3!}\prod_{j=1}^{d-1}n_j! + \binom{n_k-2}{2}\frac{1}{2!^2}\prod_{j=1}^{d-1}n_j!$ many 2-nd order permutation points at layer $k$.

### (3) The case $K = 3$:

There are three ways to have $n_k - 3$ distinct vectors out of $n_k$, corresponding to three unordered partitions of $n_k$: (i) $n_k = 4+1+\ldots+1$, (ii) $n_k = 3+2+1+\ldots+1$, and (iii) $n_k = 2+2+2+1+\ldots+1$.

#### (i) $n_k = 4 + 1 + \ldots + 1$
For this case, we have $\frac{n_k!}{4!}$ permutations given by permuting the neuron indices of layer $k$. Therefore, this 3-rd order permutation point induces a permutation set with cardinality $|P(\boldsymbol{\theta})| = \frac{1}{4!}\prod_{j=1}^{d-1}n_j!$.

As usual, we should consider other 3-rd order permutation points (at layer $k$) giving rise to the same network function and corresponding to the same unordered partition. If we had chosen another parameter vector to replicate four times, this 3-rd order permutation point would induce another permutation set. Note that we can choose the parameter vector to replicate out of $n_k - 3$ in $\binom{n_k-3}{1}$ ways and there are $\frac{1}{4!}\prod_{j=1}^{d-1}n_j!$ many points in each one of the permutation sets. Therefore, we end up having $\binom{n_k-3}{1}\frac{1}{4!}\prod_{j=1}^{d-1}n_j!$ many 3-rd order permutation points (at layer $k$) corresponding to this unordered partition.

#### (ii) $n_k = 3 + 2 + 1 + \ldots + 1$
For this case, we have $\frac{n_k!}{3!2!}$ permutations given by permuting the neuron indices of layer $k$. Therefore, this 3-rd order permutation point induces a permutation set with cardinality $|P(\boldsymbol{\theta})| = \frac{1}{3!2!}\prod_{j=1}^{d-1}n_j!$.

As usual, we should consider other 3-rd order permutation points (at layer $k$) giving rise to the same network function and corresponding to the same unordered partition. If we had chosen two other parameter vector to replicate, say $\boldsymbol{\vartheta}_l'^{(k)}$ and $\boldsymbol{\vartheta}_m'^{(k)}$, this 3-rd order permutation point would induce another permutation set. Note that we can choose two parameter vectors to replicate out of $n_k - 3$ in $2!\binom{n_k-3}{2}$ ways. We have an extra 2! factor here since we have different permutation sets if we replicate $\boldsymbol{\vartheta}_l'^{(k)}$ twice and $\boldsymbol{\vartheta}_m'^{(k)}$ three times, or vice versa. Yet, there are $\frac{1}{3!2!}\prod_{j=1}^{d-1}n_j!$ many points in each one of the permutation sets. Therefore, we end up having $2!\binom{n_k-3}{2}\frac{1}{3!2!}\prod_{j=1}^{d-1}n_j! = \binom{n_k-3}{2}\frac{1}{3!}\prod_{j=1}^{d-1}n_j!$ many 3rd-order permutation points (at layer $k$) corresponding to this unordered partition.

#### (iii) $n_k = 2 + 2 + 2 + 1 + \ldots + 1$
For this case, we have $\frac{n_k!}{2!^3}$ permutations given by permuting the neuron indices of layer $k$. Therefore, this 3-rd order permutation point induces a permutation set with cardinality $|P(\boldsymbol{\theta})| = \frac{1}{2!^3}\prod_{j=1}^{d-1}n_j!$.

As always, we should consider the other 3-rd order permutation points (at layer $k$) giving rise to the same network function and corresponding to the same unordered partition. Note that we can choose three parameter vectors to duplicate out of $n_k - 3$ in $\binom{n_k-3}{3}$ ways and there are $\frac{1}{2^3}\prod_{j=1}^{d-1}n_j!$ many points in each one of the permutation sets. Therefore, we end up having $\binom{n_k-3}{3}\frac{1}{2^3}\prod_{j=1}^{d-1}n_j!$ many 3-rd order permutation points (at layer $k$) corresponding to this unordered partition. Overall, we have $\binom{n_k-3}{1}\frac{1}{4!}\prod_{j=1}^{d-1}n_j! + \binom{n_k-3}{2}\frac{1}{3!}\prod_{j=1}^{d-1}n_j! + \binom{n_k-3}{3}\frac{1}{2!^3}\prod_{j=1}^{d-1}n_j!$ many 3-rd order permutation points at layer $k$.

### (4) A note on the general closed form formula for $T(K, n_k)$:

For a general integer $K$ there is no closed-form formula for the number of partitions, although it is easy to have a closed-form formula for the number of permutation points for one given unordered partition following the counting arguments. Thus, we believe that there is no way to find a closed-form formula for $T(K, n_k)$ as a function of $K$. However, the lower bound for $T(K, n_k)$ is also the dom-

inating term for large $n_k$, since every other summand would be a polynomial of at most $(K1)$-th order.

**(5) A lower bound for general $K$:**

For general $K$, we have $l = n_k - K$ distinct parameter vectors in the small network. There are many ways to partition $n_k$ into $l$ positive integers without respecting order. Since we are interested in a lower bound, we only consider the following unordered partition: $n_k = 2 + \ldots + 2 + 1 + \ldots + 1$, i.e. we have $K$ duplicated parameter vectors and $n_k - 2K$ parameter vectors that appear once. For this unordered partition, we have $\binom{n_k - K}{K}$ ways to choose the duplicated parameter vectors. For each one of these choices, we can permute the neuron indices in $\frac{n_k!}{2^K}$ different ways. Including the permutations in other layers $j \neq k$, we end up with $\binom{n_k - K}{K} \frac{1}{2^K} \prod_{j=1}^{d-1} n_j!$ points in the permutation set. The number is a lower bound of $T(K, n_k)$, because other unordered partitions of $n_k$ give rise to other $K^{\text{th}}$-order permutation points at layer $k$.

## C.5 PROOF OF LEMMA 2

We can approximate the factorial an integer $n$ using Stirling's formula

$$n! \to \sqrt{2\pi n}\left(\frac{n}{e}\right)^n \text{ as } n \to \infty \tag{4}$$

We note that this approximation leads to accurate results even for small $n$.

As $n \to \infty$, both $n - K \to \infty$ and $n - 2K \to \infty$ for finite $K$. Therefore, we can apply Stirling's formula both for $n - K$ and $n - 2K$:

$$\frac{1}{2!^K}\binom{n-K}{K} = \frac{1}{2!^K}\frac{(n-K)!}{(n-2K)!K!} \tag{5}$$

$$\frac{1}{2!^K}\frac{(n-K)!}{(n-2K)!K!} \to \frac{1}{2!^K}\frac{\sqrt{2\pi(n-K)}\left(\frac{n-K}{e}\right)^{n-K}}{\sqrt{2\pi(n-2K)}\left(\frac{n-2K}{e}\right)^{n-2K}K!} \text{ as } n \to \infty \tag{6}$$

$$= \frac{1}{2!^K}\sqrt{\frac{n-K}{n-2K}}\frac{(n-K)^{n-K}}{(n-2K)^{n-2K}}\frac{1}{e^K K!} \text{ as } n \to \infty \tag{7}$$

$$= \frac{1}{2!^K}\sqrt{\frac{\frac{n}{K}-1}{\frac{n}{K}-2}}\frac{(\frac{n}{K}-1)^{n-K}}{(\frac{n}{K}-2)^{n-2K}}K^K\frac{1}{e^K K!} \to \frac{1}{2!^K}(\frac{n}{K})^K\frac{K^K}{e^K K!} \text{ as } n \to \infty \tag{8}$$

$$= \frac{1}{2!^K}\frac{1}{e^K K!}n^K = c_K n^K \text{ as } n \to \infty \tag{9}$$

## C.6 PROOF OF LEMMA 3

We already know the number of equilavent $K$-th order permutation points (the ones that reduce to the same configuration in the landscape of the neural network with $(n_1, \ldots, n_k - K, \ldots, n_d)$) is at least $\sum_{k=1}^{d-1} T(K, n_k) \prod_{k=1}^{d-1} n_k!$ (Proposition 3-(ii)). We can easily that see that every permutation point gives rise to a *distinct* high-dimensional subspace of critical points by observing that their parameter vectors in layer $k$ would be in distinct positions when projected on the parameters of the layer $k$ since we never repeat the same set of parameters in layer $k$. If two such subspaces were the same, their projection on a lower dimensional space (parameters of the layer $k$) would be same necessarily. Therefore, the subspaces mentioned are distinct and the number of them is equivalent to the number of related permutation po

