# OpenReview forum: "Weight-space symmetry in neural network loss landscapes revisited"
_ICLR.cc/2020/Conference — Reject_

### Official Review · AnonReviewer3 · 2019-10-22
**Official Blind Review #3**

**Rating:** 3

**Review:**

The paper presented a method for studying the landscape of the loss function w.r.t. parameters in a neural network from the perspective of weight-space symmetry. The detailed method includes constructing/optimising a low-loss path in between two parameter vectors (incoming connections from the previous layer) of two neurons respectively, and then set the output weight vectors (outgoing connections to the next layer) to be the same without changing the output of the current layer.

The empirical results show that the proposed optimisation scheme for finding the path is indeed low-loss, which implies that there exists numerous critical points in between two equivalent local minima (which are global minima in over-parametrised models).

My major concern is that scope of the study is very limited, since only permutation is considered here. I am not an expert in this field, however, I could come up with examples which could easily generalise the method of study to rotations since rotation matrices include permutation matrices.

We can look into the loss landscape of 2D matrix factorisation in this form UV^T=W, and it is obvious that any rotation of U on the RHS of it is an optimal solution as long as the same rotation is applied to V. The perspective from this optimisation problem is that it gives us a continuous plateau and it includes permutation of the dimensions.

For neural networks with only one hidden layer. For example, consider a neural network in this form y=Uf(Vx) where U and V are parameter matrices and f() is a monotonic squashing function. It is easy to show that, when U is timed with a matrix R, where RR^T= I, as in U'= UR, there exist a V' and \alpha that gives R^Tf(Vx) = \alpha f(V'x) so that Uf(Vx) = URR^Tf(Vx) = U'f(V'x) for the hyperbolic tangent function. In the case where ReLU activation function is used as f(), then as long as the rotation doesn't produce negative entries, V' exists. In addition, when f is ReLU, isotropic scaling of the outputs from the first layer also gives rise to equivalent optima.

Compared to the proposed study, the aforementioned way of studying the neural networks naturally gives continuous plateaus w.r.t. U in the loss landscape, and, by studying the discontinuity of the landscape w.r.t. V, more understanding could be unveiled.

**Experience Assessment:**

I have read many papers in this area.

**Review Assessment: Checking Correctness Of Derivations And Theory:**

I carefully checked the derivations and theory.

**Review Assessment: Checking Correctness Of Experiments:**

I carefully checked the experiments.

**Review Assessment: Thoroughness In Paper Reading:**

I read the paper at least twice and used my best judgement in assessing the paper.

---

> ### Author Response · Authors · 2019-11-14
> **Response to Reviewer 3**
>
> Thank you for your feedback and ideas.
>
> The symmetries you describe for linear networks are well-known in the field (see e.g. Baldi and Hornik 1989 https://doi.org/10.1016/0893-6080(89)90014-2 ) and generalizations to non-linear activation functions are possible thanks to local linear approximations. That is, it may be that (local) minima are embedded in small flat subspaces that can be characterized with such symmetries. It is, however, for networks with non-linear activation functions in general not possible to move along an equal-loss path from one permutation of the hidden neurons to another permutation (because the linear approximations hold only locally). Our work shows one way to find nevertheless low-loss paths between such permutations in non-linear neural networks.

---

### Official Review · AnonReviewer2 · 2019-10-23
**Official Blind Review #2**

**Rating:** 6

**Review:**

This paper studies the permutation symmetry of deep neural networks. It was known that by reordering neurons and their connections in each layer, the input -> output map the neural network represents can be preserved. This corresponded to a set of unconnected equivalent points in the weight space. The authors study the weight-space connectivity of these points through midpoints they call “permutation points”. They demonstrate that such points are members of high-dimensional manifolds of equivalent points. After that, they look at empirical experiments and explicitly construct a path between two equivalent weight-space points on a toy task and MNIST.

I generally like the paper and its geometrical lens on the problem. I find the figures very helpful in understanding what is going on. However, there are a few points that I wasn’t entirely clear on. I will detail them below.

-- Point 1 --
Connecting equivalent minima vs connecting SGD-found minima.

If I understood the paper correctly, the derivation connects two weight space points A and B whose weights and biases, once loaded to the neural network, would have the exact same answers on all inputs X i.e. f_A(X) == f_B(X), i.e. they are a pair of equivalent points. I understand that those are the ones we obtain by using the permutation symmetries.

However, some of the papers cited look at the low-loss paths between pairs of optima found by training with SGD from independent initializations, which in turn represent different functions. I.e. for two such optima C and D, the predictions on the val/test set are (sometimes) different, showing that the functions are not the same. I found the initial evidence in:

Large Scale Structure of Neural Network Loss Landscapes. Stanislav Fort and Stanislaw Jastrzebski. NeurIPS 2019. (https://arxiv.org/abs/1906.04724)

And also in much more detail in another OpenReview submission:

Deep Ensembles: A Loss Landscape Perspective. (https://openreview.net/forum?id=r1xZAkrFPr)

I found your results very compelling, however, the two problems seem to be quite different -- on one hand you are connecting a pair of minima that are in fact *identical* by construction. On the other hand the empirical work in literature (especially in the two papers I provided above) deals with pairs of minima that in fact do differ on the test set (at least).

Would you mind commenting on how the two approaches relate to each other?

-- Point 2 --
Higher order connectivity

In Large Scale Structure of Neural Network Loss Landscapes. Stanislav Fort and Stanislaw Jastrzebski. NeurIPS 2019. (https://arxiv.org/abs/1906.04724), the authors look at higher-order connectivity between SGD-found optima (e.g. connecting 3 optima on a 2-manifold, 4 optima on a 3-manifold etc.). They also have a particularly simple path-finding algorithm. This seems relevant to the approach you are presenting, although the points in Point 1 still stand.

-- Point 3 --
Previous work on connecting two optima using layer-wise weight merging

Explaining Landscape Connectivity of Low-cost Solutions for Multilayer Nets. Rohith Kuditipudi, Xiang Wang, Holden Lee, Yi Zhang, Zhiyuan Li, Wei Hu, Sanjeev Arora, Rong Ge. (https://arxiv.org/abs/1906.06247)

They prove that a low-loss path between 2 optima exists provided you can apply a p_keep = 0.5 dropout on each of the optima without incurring a significant loss punishment for it. This paper seems very related to your approach. Would you mind commenting on the differences?

-- Conclusion --
I like the paper and the idea in general. I appreciate the geometrical lens the authors took. My main point of confusion relates to the connection between this work and the low-loss connectivity between inequivalent optima found in literature, which (at least to me) seems to be the more interesting of the two connectivities.


**Experience Assessment:**

I have published one or two papers in this area.

**Review Assessment: Checking Correctness Of Derivations And Theory:**

I assessed the sensibility of the derivations and theory.

**Review Assessment: Checking Correctness Of Experiments:**

I assessed the sensibility of the experiments.

**Review Assessment: Thoroughness In Paper Reading:**

I read the paper at least twice and used my best judgement in assessing the paper.

---

> ### Author Response · Authors · 2019-11-14
> **Response to Reviewer 2**
>
> Thank you for your thorough feedback and for pointing to some interesting references.
>
> Connecting ‘partner’ minima vs. connecting independent minima: Indeed, our pathfinding method can only link ‘partner’ minima whereas other methods can find paths between independent minima (representing different functions). Distinctively, our method enables observing how individual weight vectors move from a minimum to a permutation point (see Fig 2) and equivalently how saddles emerge in between minima, therefore provides a geometric perspective on the landscape.
>
> Relation to Fort et al. 2019: Their work introduces a method that enables connecting m minima through an (m+1)-dim manifold. In our work, we prove that permutation points lie in high-dimensional plateaus of critical points (Lemma 3). In other words, one can pick any m critical points (or infinitely many) in this hyperplane and these points are connected in an n_k-dim hyperplane (n_k is the number of neurons in the next layer, this is usually bigger than 10, thus suggesting high-dimensional manifolds compared to the empirical findings in Fort 2019 for m=10). There is also an interesting connection between these two works: in their phenomenological model, the wedges intersect, suggesting connectivity of minima (see their Fig1) whereas we prove that the permutation points are connected at equal-loss (see our Fig 4 in the appendix).
>
> A lower bound on the number of critical points: We also wanted to point out to an interesting part in our paper: in addition to studying landscape connectivity, we find a lower bound for the number of permutation points corresponding to the midpoints in our pathfinding algorithm. We prove that there are at least polynomially more permutation points than the global minima (Proposition 3).
>
> Relation to Kuditipudi et al. 2019: Their method seems quite powerful while being quite general. However, their requirement of robustness against a 50% dropout, although practically no problem for big networks, is a major assumption and could actually be difficult to meet in layers with only a few units (e.g. 32 filters in an early convolutional layer).

---

### Official Review · AnonReviewer1 · 2019-10-26
**Official Blind Review #1**

**Rating:** 3

**Review:**

This paper studies a special type of weight symmetry in neural networks. I think studying the geometry of neural-nets is an interesting and important direction for understanding neural-nets, and along this direction, weight-space symmetry is an important subject. However, it seems to me this paper does not make enough contributions. Details are given below.

List of contributions of the paper:

1.	Propose an algorithm to find a (low-loss) path connecting arbitrary two partner local minima and passing through a permutation point, where a permutation point is defined as a weight setting where a pair of neurons (in the same layer) have the same fan-in and fan-out weights.

2.	Theoretically prove that some permutation points are connected via paths with equal loss. (Proposition 1 and 2)

3.	Provide a lower bound for the number of permutation points and high-order permutation points (Proposition 3).

Cons:

1.	The theory of this paper is a bit weak. There are three propositions.  Proposition 1 and Proposition 2 are about the equal-loss surface and theoretical existence of an equal-value path. They are kind of straightforward to prove. Prop. 3 is about counting the number of permutation points. It is a rather simple combinatorial problem, and the lower bound of the expression (an exponential bound) seems standard.

2.	Simple proofs can sometimes provide nice insight, but it is not clear how the study of partner global minima can help improve the understanding of DNN.
    --First, partner global minima are just a special case of critical points created by "neuron splitting", which has been comprehensively studied in  [FA2000] (Fukumizu and Amari, 2000). Note that Theorem 1 of this paper is also directly borrowed from [FA2000]. To me, this paper does not provide much additional theoretical insight on neuron splitting.
    --Second, what is the significance of the simulation results? Prior works have shown the existence of low-cost path; this paper shows the existence of a low-cost path containing a permutation point. The major difference is that the new low-cost path is more special. Why is this finding interesting and useful? (noting that the proof of the existence of such a path seems to be much easier than proving the existence of such a path for two general global minima).


3.	It is not clear how the path-finding algorithm helps in practice.
a)	The algorithm is computationally expensive. It is a double-loop algorithm: in the outer loop, d is reduced by a tiny amount at each time; in the inner loop, for a fixed d, gradient descent is run for the entire DNN (with only one parameter excluded) until convergence. The total time is (# of d) * (original running time). Here, # of d’s depends on the grid: if the initial d is 10 and the grid size is 0.1, then (# of d) = 100.
     Compared to the path-finding algorithm in Garipov et al., which only runs GD for one time, this algorithm is much more expensive, yet the benefit is unclear.
b)	The motivation of the algorithm is not clear. Why choose to monotonically decrease the difference between the fan-in weights (of the chosen pair of neurons), but let all other weights freely optimized during the pathfinding algorithm? The paper does not provide a detailed explanation of this.

4.	Minor issues
  -- The weight parameters created by Algorithm 1 may not be a critical point of the loss function, and thus not necessarily a saddle point. I think the claim “since the path connects two partner minima, there must be at least one saddle point on the path” is incorrect without extra assumptions.
  --The paper mentioned "global minima" in the introduction; but in practical training, one does not always find global minima. Is "global minima" crucial for the theory and for the algorithm?

**Experience Assessment:**

I have published one or two papers in this area.

**Review Assessment: Checking Correctness Of Derivations And Theory:**

I assessed the sensibility of the derivations and theory.

**Review Assessment: Checking Correctness Of Experiments:**

I assessed the sensibility of the experiments.

**Review Assessment: Thoroughness In Paper Reading:**

I read the paper at least twice and used my best judgement in assessing the paper.

---

> ### Author Response · Authors · 2019-11-14
> **Response to Reviewer 1**
>
> Thank you for your thorough feedback.
>
> Related to Proposition 3: We have not encountered other work in the literature that gives a lower bound on the number of critical points (that have higher loss than the global minimum). We see our contribution in proving that there are polynomially more permutation points than global minima in the landscapes of multi-layer neural networks (Proposition 3). Furthermore, we prove that these permutation points lie in high-dimensional plateaus of critical points (Lemma 3).
>
> A new insight on “neuron splitting”: In addition to reviewing the results from Fukumizu and Amari, we prove that ALL permutation points are connected through continuous paths of the same loss.
>
> Relation to Garipov et al.: In Garipov et al the authors search through certain parametrized paths to connect independent minima (specifically, polygonal chains and bezier curves) and find optimal parameters that yield the lowest barrier (for a chosen parametrization). In our pathfinding method, we do not restrict our path method to a certain geometrical family. This enables finding paths of arbitrary shape and thus may lead to lower-barrier paths at the cost of connecting only the ‘partner’ minima.
>
> Saddles between global minima: We agree that the weight parameters (points on the path) created by our Algorithm 1 may not be a critical point of the loss function. However there must be at least one saddle point on the path as can be seen as follows: Either the permutation point is already a saddle or it is a local minimum of the loss. In the second case, the path must cross a point of higher loss at some point theta(d_s). By construction, at every point on the path, the derivative is zero in all directions but in d-direction, i.e. in the direction of the path. Since the loss values on the path before and after the point theta(d_s) are lower, the derivative at theta(d_s) in d-direction (along the path) must be zero as well. Thus theta(d_s) is a critical point; the saddle we were looking for.
>
> Connecting the “global minima”: In the paper we use a teacher-student setup to ensure being in a global minimum (of the student loss) before applying Algorithm 1. However we expect to find similar results for paths connecting local minima as encountered in classical network training and experiments on this are currently running.

---

### Public Comment · ~Micah_Goldblum1 · 2019-11-08
**An Interesting Connection**

Hi Authors,
Thank you for your interesting paper.  I noticed that your work concerning minima of the loss landscape is related to our paper, which yields both theoretical and empirical results concerning the suboptimal local minima.[1]  Please consider mentioning the relationship with our work in your next version.

[1] https://arxiv.org/abs/1910.00359

---

> ### Author Response · Authors · 2019-11-14
> **Response**
>
> Thank you for pointing to your interesting work. We will cite your work where appropriate in the next version.

---

### Decision · Program_Chairs · 2019-12-19

**Decision:**

Reject

**Comment:**

After communicating with each reviewer about the rebuttal, there seems to be a consensus that the paper contains a number of interesting ideas, but the motivation for the paper and the relationship to the literature needs to be expanded.  The reviewers have not changed their scores, and so there is not currently enough support to accept this paper.